# Arene C–H borylation strategy enabled by a non-classical boron cluster-based electrophile

Sangmin Kim [1], Joseph W. Treacy[1], Yessica A. Nelson [1], Jordan A. M. Gonzalez[1], Milan Gembicky [2], K. N. Houk [1] & Alexander M. Spokoyny [1,3] ✉

Introducing a tri-coordinate boron-based functional group (e.g., boronic ester) into an unactivated C–H bond in the absence of directing groups is an ongoing challenge in synthetic chemistry. Despite previous developments in transition metal-catalyzed and -free approaches, C–H borylation of sterically hindered arenes remains a largely unsolved problem to date. Here, we report a synthetic strategy of a two-step, precious metal-free electrophilic C–H borylation of sterically hindered alkyl- and haloarenes to generate aryl boronic esters. The first step relies on electrophilic aromatic substitution (EAS) induced by cage-opening of $Cs_2[closo\text{-}B_{10}H_{10}]$, forming a $6\text{-}Ar\text{-}nido\text{-}B_{10}H_{13}$ product containing a B–C bond, followed by a cage deconstruction of arylated decaboranes promoted by diols. The combination of these two steps allows for the preparation of aryl boronic esters that are hardly accessible by current direct C–H borylation approaches. This reaction does not require any precious metals, highly-engineered ligands, pre-functionalized boron reagents, or inert conditions. In addition, the unique properties of a non-classical boron cluster electrophile intermediate, $B_{10}H_{13}{}^+$, afford a regioselectivity with unique steric and electronic control without the undesirable side reactions.

Organoboron compounds are now considered one of the most essential building blocks for synthesis[1–4]. Among these, organic molecules with $C(sp^2)$–B bonds represent a large and important class of substrates that can be transformed into various derivatives through a multitude of straightforward approaches relying on the replacement of the tri-coordinate boron-based moiety[5,6]. As a general approach toward diverse boron-based functional groups from their simple pre-functionalized precursors, direct borylation of $C(sp^2)$–H bonds utilizing transition metal catalysts[7–9] or electrophilic aromatic substitution (EAS)[10] have been previously studied with emphases on both reactivity and selectivity (Fig. 1a). For transition metal catalysis, various strategies to gain specific regioselectivity (*ortho* vs. *meta* vs. *para*) such as introduction of directing groups[11–13] or utilization of steric[14–17] or electronic effects[18–20] resulting from the pre-installed functional groups have been

extensively developed. In particular, alkylbenzenes can be borylated under sterically controlled catalytic systems which give *meta* selectivity to alkyl groups[14–17]. However, the steric control can also limit the scope of alkylbenzenes because the catalyst cannot access sterically hindered C–H bonds, notably those *ortho* to alkyl groups. Consequently, transition-metal catalyzed borylation of sterically hindered arenes such as mesitylene where all aromatic C–H bonds are *ortho* to alkyls has been limited[21,22]. An exceptional example was reported by Chatani et al. utilizing an NHC-ligated platinum catalyst, [(ICy)Pt(dvtms)] (ICy = 1,3-dicyclohexylimidazol-2-ylidene, dvtms = divinyltetramethyldisiloxane) where sterically encumbered alkylbenzenes such as mesitylene could be borylated to give corresponding aryl boronic esters[21].

As an alternative to transition metal-catalyzed chemistry, electrophilic arene borylation using boron-based electrophiles follows the

[1]Department of Chemistry and Biochemistry, University of California, Los Angeles, Los Angeles, CA 90095, USA. [2]Department of Chemistry and Biochemistry, University of California, San Diego, La Jolla, CA 92093, USA. [3]California NanoSystems Institute, University of California, Los Angeles, Los Angeles, CA 90095, USA. ✉e-mail: spokoyny@chem.ucla.edu

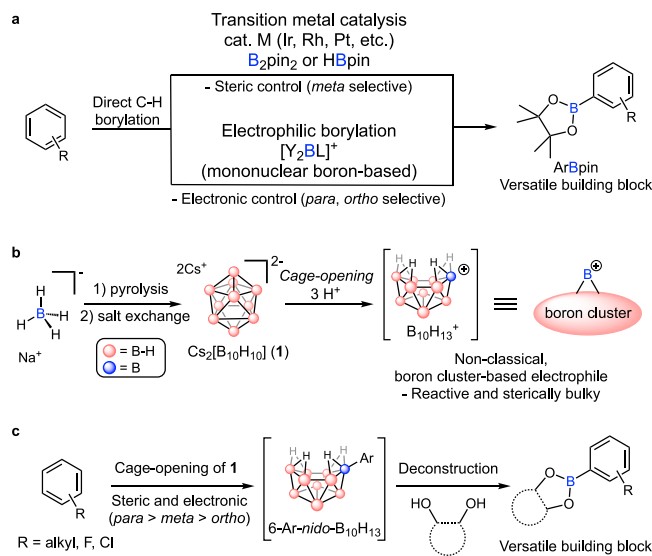

**Fig. 1 | Borylation of aromatic C–H bonds. a** Conventional routes to aryl boronic esters by direct arene borylation. **b** Cage opening of $[B_{10}H_{10}]^{2-}$ to generate a non-classical boron cluster-based electrophile. **c** Synthetic approach through cage opening of $[B_{10}H_{10}]^{2-}$ followed by the deconstruction of the corresponding 6-Ar-*nido*-$B_{10}H_{13}$ species.

general trend of reactivity and selectivity of EAS mechanism and thus shows selectivity in electron-rich C–H bonds (electronic control), often at the *ortho*- or *para*-positions of alkylbenzenes (Fig. 1a)[10]. In general, classical boron-based electrophiles consist of a single boron moiety connected to auxiliary mono- or polydentate ligands that control the electrophilicity of the boron center. However, the intrinsically weak electrophilicity of the currently known mononuclear boron-based electrophiles significantly confines the scope of their reactivity to highly electron-rich heteroarenes or anilines[23–27]. Some notable exceptions reported by Ingleson utilizing in situ generated $[Y_2BL]^+$ ($Y_2$ = catecholato, $Cl_2$; L = amine) as mononuclear boron-based electrophiles have shown activity toward alkylbenzene substrates[28–31]. However, these systems require high reaction temperatures (>70 °C) due to insufficient electrophilicity of the generated borenium intermediates. The high reaction temperature results in poor regioselectivity as well as undesirable alkyl shift or elimination reactions originating from the Wheland (arenium) intermediates[32–34]. Furthermore, these reactions normally necessitate rigorously inert conditions due to the high air and moisture sensitivity of the reagents employed. To overcome these limitations, stronger boron electrophiles that can be generated from bench-stable and simple precursors are highly desired.

In this work, we now report a strategy for C(sp²)–H borylation relying on the use of non-classical boron cluster-based electrophiles which are significantly more reactive than their mononuclear congeners (see above). Hawthorne has previously shown an acid-induced cage-opening reaction of dicesium *closo*-decaborate ($Cs_2[B_{10}H_{10}]$, **1**) that forms an intermediate capable of reacting with several arenes and generating 6-Ar-*nido*-$B_{10}H_{13}$ as the products, which represents a unique example of a transition-metal-free cluster-based borenium species (Fig. 1b)[35–37]. Importantly, this polyhedral borane **1** can be obtained in one step from thermolysis of sodium borohydride and is air and moisture stable. We hypothesized that an acid-generated non-classical, strongly electrophilic boron cluster intermediate, $B_{10}H_{13}^+$, would be suitable for a broad scope of electrophilic aromatic borylation of various densely substituted arenes. We further hypothesized that the resulting arylated *nido*-decaborane compounds could be converted into simple Bpin-containing products via a selective cage deconstruction which allows the breakage of B–B bonds while

preserving B–C bonds. While large boron clusters such as $[B_{10}H_{10}]^{2-}$ are generally thought to be very chemically inert entities[38–41], we recently discovered that a selective cage deconstruction is feasible in the context of $[B_6H_5R]^{2-}$ (R = alkyl) species, providing a potential pathway for converting multinuclear boron clusters into mononuclear congeners[42]. Here, we show that combining these two synthetic steps provides a precious metal- and ligand-free strategy for the preparation of aryl boronic esters from **1** and alkyl- and haloarenes by sequential cage opening of **1** and the deconstruction of the resulting boron cages (Fig. 1c).

## Results and discussion

In order to evaluate the feasibility of our proposed strategy, we have prepared 6-Tol-*nido*-$B_{10}H_{13}$ (**2**) (Tol, tolyl) as a model substrate by reacting **1** and 5 equivalents of HOTf in toluene[36]. The resulting product was purified by silica gel column chromatography and isolated in 89% yield as a white, air-stable solid characterized by $^1H$, $^{13}C$ and $^{11}B$ NMR spectroscopy. To evaluate whether **2** can undergo cage deconstruction, we commenced with an NMR spectroscopic study to examine the potential conversion of **2** into a mono-boron species in the presence of pinacol (Fig. 2). We hypothesized that these conditions could be feasible, considering our previous study which showed that *nido*-$B_5R_5$ (R = alkyl) species could undergo deconstruction forming RBpin products[42]. First, in a J. Young NMR tube, a $CD_3CN$ solution of **2** (*p*:*m* = 72:28) was prepared under a static vacuum (Fig. 2b, c, red line). The solution was heated at 65 °C for 24 h, but no conversion of **2** was observed, showing high thermal stability of **2** in acetonitrile (Fig. 2b, c, olive line). Next, 5 equivalents of pinacol were added to the $CD_3CN$ solution of **2**. When the solution was heated at 65 °C for 24 h, formation of TolBpin (**3**) and gaseous $H_2$ was observed by $^1H$ NMR spectroscopy along with a decreased concentration of **2** (Fig. 2b, green line). Also, $^{11}B$ NMR spectrum of this solution revealed the formation of HBpin as the byproduct (Fig. 2c, green line). Further heating of the solution for a total of 72 h achieved a nearly quantitative conversion of **2** and HBpin, successfully yielding **3** as the only observable major organic species by NMR spectroscopy (Fig. 2b, c, purple line). The ratio of *para*- and *meta*-isomers of **3** were 73:27 almost identical to those of the starting **2**, which indicates that rearrangement of the C(Tol)-B bond does not take place during the reaction. Although the detailed reaction mechanism is currently elusive due to a lack of observable core intermediates other than HBpin, the formation of $H_2$ indicates similarity to the reactions of *nido*-$B_{10}H_{14}$ with small nucleophiles, where $H_2$ is formed as the major byproduct and thus implies coordination of pinacol to the boron cage is likely to occur during the reaction[43–45].

With the initial observation of the formation of aryl boronic ester in hand, optimization experiments were conducted using in situ generated 6-Ph-*nido*-$B_{10}H_{13}$ as a model substrate to generate PhBpin (**4**) as a target product (Table 1). The model substrate, 6-Ph-*nido*-$B_{10}H_{13}$ was generated from **1** and benzene at room temperature for 3 h under aerobic conditions and then used in situ after simple filtration through a silica plug without thorough isolation. For the deconstruction step, a reaction mixture of in situ generated 6-Ph-*nido*-$B_{10}H_{13}$ with 10 equivalents of pinacol and 12 equivalents of $MgSO_4$ as drying reagent in MeCN was stirred at 65 °C for 24 h under air conditions. Without any deviations from these conditions, **4** was produced in 24% NMR spectroscopic yield along with 40% of the remaining 6-Ph-*nido*-$B_{10}H_{13}$ determined by $^1H$ NMR spectroscopy using mesitylene as an internal standard (entry 1). The addition of 10 mol% *p*-BQ (1,4-benzoquinone) as an oxidant did not affect the reaction efficiency, unlike the previous hexaborate degradation where superstoichiometric amounts of oxidants were needed (entry 2)[42]. However, the use of 10 mol% of a stronger oxidant, CAN (cerium(IV) ammonium nitrate), increased the reaction yield to 48% and only 9% of **2** remained after 24 h (entry 3). To complete the screening, increased loading (20 mol%) of CAN was examined, and it

produced **4** in 71% NMR spectroscopic yield with the full conversion of 6-Ph-*nido*-$B_{10}H_{13}$ (entry 4). When the reaction was performed under an $N_2$ atmosphere, a slightly decreased yield of **4** was obtained, but the difference was not significant (entry 5). Because the reaction can proceed without an oxidant, we postulate that the role of CAN is to facilitate the oxidation of boranes followed by the coordination of pinacol[46].

With the optimized conditions, various alkyl- and halobenzenes were examined (Fig. 3a). The target compound of the screening experiment, **4**, was isolated in 67% yield after purification of the

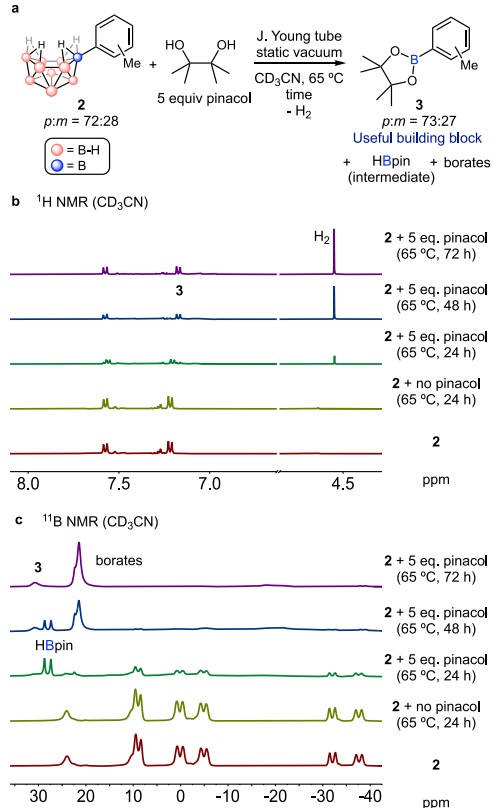

**Fig. 2 | Monitoring of deconstruction of 2 by NMR spectroscopy. a** Formation of **3**, $H_2$, HBpin and borates by the deconstruction of **2** in the presence of pinacol in $CD_3CN$. **b** Stacked $^1H$ NMR spectra of the reaction mixture. **c** Stacked $^{11}B$ NMR spectra of the reaction mixture.

reaction mixture by silica gel flash column chromatography. Mono-substituted benzene compounds gave up to 83% yields of the corresponding aryl boronic esters (**3,5–9**) with high *para* selectivity following the electronic control commonly observed in EAS reactions. Sterically bulkier alkyl groups showed higher *para* selectivity in general. Next, alkylbenzenes that do not have available *para* C–H bonds were investigated. For 1,2-disubstituted benzene derivatives, substitutions at the 4-positions were obtained like the previously reported electrophilic borylations (**10–12**)[28]. However, very unusual selectivity was observed for 1,3-disubstituted benzenes such as *m*-xylene. Unlike the previous electrophilic borylation examples which showed selectivity for 4-positions (*ortho* to alkyl groups)[28], substitution at 5-positions (*meta* to alkyl groups) was obtained with very high regioselectivity (**13–15**). As all the previously reported *meta*-selective borylation of alkylbenzenes were proposed to follow the steric control[7–9,14–17], this unusual *meta* selectivity also could result from the steric effect caused by the bulkiness of the non-classical boron cluster electrophile. Next, alkylbenzene compounds that have only *ortho* C–H bonds were examined. Previous electrophilic borylation reactions which were typically conducted at high temperature (>70 °C) usually did not work with these substrates and even a few known examples exclusively showed undesirable side reactions such as alkyl migration and elimination[28–31]. However, our reaction conditions showed good reactivity with *p*-xylene resulting in a 79% yield of **16** without any side reactions. To further investigate the steric effect in this system, 1,4-dialkylbenzenes with two different alkyl groups were tested. Consequently, the less sterically hindered C–H bonds *ortho* to methyl groups were exclusively borylated instead of the sterically encumbered C–H bonds *ortho* to isopropyl or *tert*-butyl groups (**17**, **18**), supporting the high steric effect of this system. Next, tri- or even more alkylated benzene derivatives such as mesitylene, 1,3,5-triethyl-, 1,2,4,5-tetramethyl- and penta-methylbenzene were examined. These substrates also showed good reactivity and the reactions did not give any undesirable side products (**20–23**), likely due to the low reaction temperature insufficient to promote alkyl shifts. It should be noted that direct C–H borylation of these sterically hindered multi-alkylbenzenes is challenging and hardly be obtained at room temperature by other methods[21].

While borenium chemistry is fundamentally incompatible with molecules containing coordinating groups (e.g., N, O) due to a strong thermodynamic propensity for the formation of strong B–N and B–O bonds, we hypothesized that borylation of haloar-enes should nevertheless be feasible[47]. To probe this, several fluoro- and chlorobenzene compounds were investigated

## Table 1 | Screening experiments

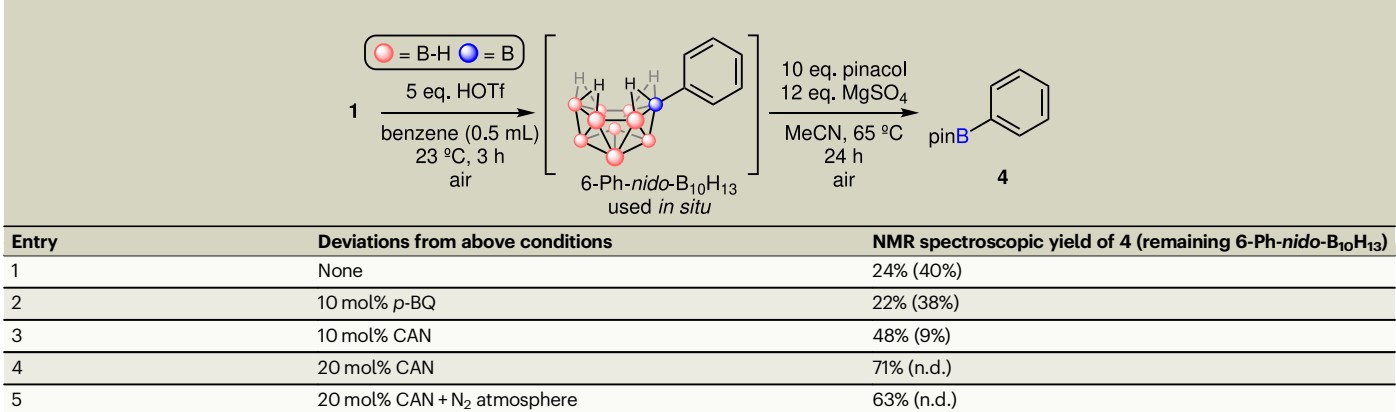

| Entry | Deviations from above conditions | NMR spectroscopic yield of 4 (remaining 6-Ph-*nido*-$B_{10}H_{13}$) |
|---|---|---|
| 1 | None | 24% (40%) |
| 2 | 10 mol% *p*-BQ | 22% (38%) |
| 3 | 10 mol% CAN | 48% (9%) |
| 4 | 20 mol% CAN | 71% (n.d.) |
| 5 | 20 mol% CAN + $N_2$ atmosphere | 63% (n.d.) |

*p*-BQ *p*-benzoquinone, *CAN* cerium ammonium nitrate, *n.d.* not detected.

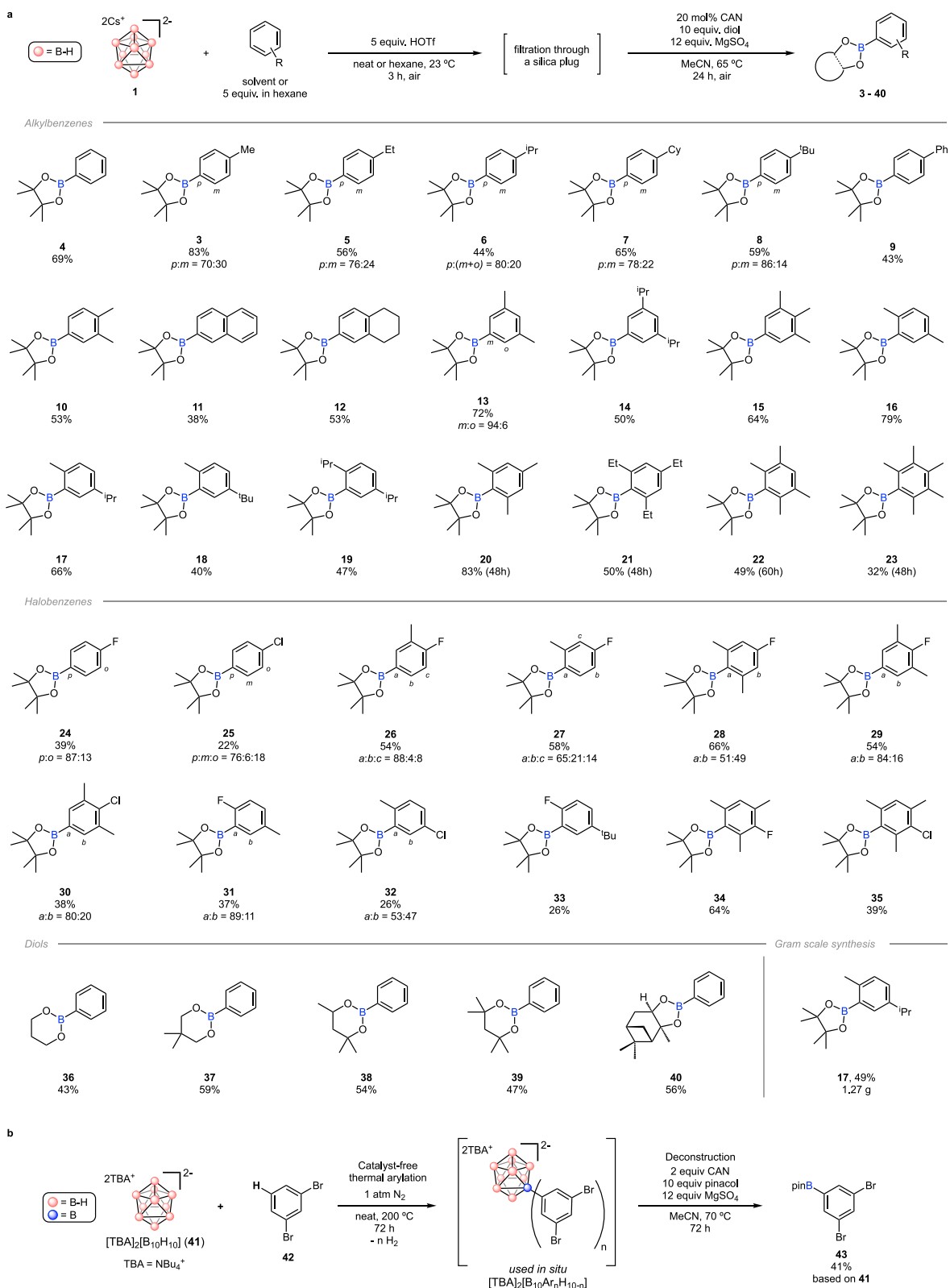

**Fig. 3 | Substrate scopes. a** Substrate scopes of the boron cluster-mediated arene C−H borylation. Longer reaction times in the deconstruction step are shown in parentheses. **b** Alternative strategy to borylate a highly electrophilic 1,3-dibromobenzene using $[B_{10}H_{10}]^{2-}$ as the boron source.

(Fig. 3a). Fluoro- and chlorobenzene gave the desired boronic ester products with *para* selectivity (**24**, **25**), but the overall yields were decreased compared to alkylbenzene substrates due to the deactivating effect of halogens in EAS[36]. When both fluoro and alkyl groups were present, the strong *para*-directing effect of

fluorine overrode the electronic and steric effects of alkyl groups in general (**26**, **27**, **31**). However, in the case where the *para*-position to fluorine is sterically hindered by two methyl groups, relatively poorer selectivity was obtained due to the steric hindrance by the methyl groups (**28**). When a C−H bond *para* to

fluorine is not available, selectivity was observed *ortho* to fluorine (**31**, **33**). In the case where all *para* and *ortho* C–H bonds to halogens are not present, borylation *meta* to halogen C–H bonds was obtained without undesirable side products (**34**, **35**). Chlorobenzenes generally showed lower selectivity and reactivity compared to fluorobenzenes when substrates have the same substitution patterns (**30**, **32**, **35**). Overall, these studies suggest that $B_{10}H_{13}^+$ intermediate can be compatible with aromatic compounds containing C–Cl and C–F bonds.

Although pinacol boronic ester (Bpin) is the most popular and desired boronic ester group in organic synthesis, other alcohol-derived boronic esters were also prepared to highlight the versatility of our borylation reaction (Fig. 3a)[4,48]. Trimethylene glycol, neopentyl glycol, hexylene glycol, 2,4-dimethyl-2,4-pentanediol, and (*1S*,*2S*,*3R*,*5S*)-(+)−2,3-pinanediol yielded the corresponding phenyl boronic esters under the standard reaction conditions (**36**–**40**). This result shows that various types of boronic esters can be prepared in a divergent manner using bare alcohols, which can be advantageous over the current borylation reactions that require multi-step syntheses to prepare pre-functionalized boron reagents such as diboron compounds[49].

A gram scale synthesis of **17** was carried out to demonstrate the scalability of this process (Fig. 3a). From 10.0 mmol of **1** in 20 mL *p*-cymene, 1.27 g of **17** was produced in a 49% isolated yield over two steps. Overall, the borylation reaction could be performed under fully aerobic conditions on a gram scale without the use of pre-functionalized boron reagents or precious metal catalysts along with unique regioselectivity and reactivity.

Despite successful C–H borylation of sterically congested alkyl-, fluoro- and chloroarenes, other types of substrates such as heteroarenes, aryl bromides or $CF_3$-substituted arenes were not suitable for our reaction conditions for two reasons. First, the use of the superacid, HOTf, resulted in rapid salt formation with most heteroarenes instead of inducing the cage opening of **1**. Also, a deleterious side reaction, formation of *nido*-$B_{10}H_{13}OTf$[36], was dominant when electron-deficient arenes such as aryl bromides and $CF_3$-substituted arenes were used. Therefore, an alternative approach was sought after to supplement these limitations and to enable **1** as a more general boron reagent for arene C–H borylation. As a result, we adopted a strategy of catalyst-free thermal arylation of *closo*-dodecaborate ($[B_{12}H_{12}]^{2-}$) where various arenes including 1,3-dibromobenzene were used as substrates to generate polyarylated dodecaborates by direct C–H bond functionalization and authors suggested that these processes occur via EAS-type mechanism[50,51]. In our attempt to apply this reaction to *closo*-decaborate functionalization[52], heating of a $[TBA]_2[B_{10}H_{10}]$ solution (**41**, TBA = tetrabutylammonium) in 1,3-dibromobenzene (**42**) at 200 °C for 72 h resulted in full conversion of **41** giving aryl-substituted decaborates, $[TBA]_2[B_{10}Ar_nH_{10-n}]$ (Ar = 1,3-$Br_2$Ph, n = 6–7) (Fig. 3b). The crude reaction mixture was simply washed with hexane to remove excess amount of remaining **42** and further heated in the presence of 2 equivalents of CAN, 10 equivalents of pinacol and 12 equivalents of magnesium sulfate in acetonitrile at 70 °C for 72 h. Consequently, 3,5-$Br_2$PhBpin (**43**) was obtained as the final product in 41% isolated yield based on the starting **41** with exclusive regioselectivity. This example showcases that a thermal route could potentially circumvent the scope of limitations of the strong acid-mediated electrophilic borylation with **1** and further shows the generality of cage deconstruction of arylated $B_{10}$-based species. This deconstruction is reminiscent of the similar processes occurring with alkylated $B_6$-based clusters previously reported by our group[36].

Next, we conducted a deeper investigation into the borylation of a challenging substrate, 1,3,5-tri-isopropylbenzene (**44**), where all C–H bonds are sterically hindered by two adjacent, sterically bulky isopropyl groups (Fig. 4). Although C–H borylation *ortho* to one isopropyl group has been reported[21], borylation of a C–H bond between two

adjacent isopropyl groups has never been achieved before. When **44** was introduced to the standard reaction conditions, some degree of migration and elimination of an isopropyl group was observed unlike methyl- or ethyl-substituted alkylbenzene compounds, probably due to the relative stability of the resulting secondary carbocation that can form from the Wheland intermediate (Fig. 4a). Nevertheless, the major product was still the desired product, 2,4,6-tri-isopropylphenyl pinacol boronic ester (**45**) with 90% selectivity determined by $^1$H NMR spectroscopy while the alkyl migration (**46**) and elimination (**14**) products were also observed in 4% and 6% ratio, respectively. When a lower reaction temperature (0 °C) was introduced to get **45** as an exclusive product, a slightly lower yield was obtained (14%), but as expected, 94% selectivity of **45** was achieved. On the other hand, a higher reaction temperature at 60 °C decreased the selectivity of **45** to 76% along with a higher ratio of **14** in 16%. Next, the reaction of HOTf with **44** at room temperature was tested because protonation of **44** by HOTf could promote alkyl shifts to make 1,2,4-tri-isopropylbenzene or 1,3-di-isopropylbenzene[33,53], followed by borylation to generate **46** or **14** (Fig. 4b). However, it turned out that HOTf does not produce the alkyl shift products. This result suggests that the alkyl migration and elimination should proceed through the formation of a Wheland intermediate formed by electrophilic addition of the boron cluster electrophile to an *ipso* position of **44** (Int_B in Fig. 4c). Between two possible pathways depicted in Fig. 4c, *path a* generating Int_A as a Wheland intermediate is likely preferred.

Next, density functional theory (DFT) calculations were conducted to further investigate the selectivity of C–H versus C–C bond activation in borylation of **44**. All DFT calculations were carried out at the ωB97X-D/6-311+G(d,p), CPCM(*n*-hexane) level of theory. First, optimized geometries of $B_{10}H_{13}^+$ and **44** and the lowest unoccupied molecular orbital (LUMO) of $B_{10}H_{13}^+$ and highest occupied molecular orbital (HOMO) of **44** were calculated (Fig. 4d). As previously suggested by Hawthorne[35,36], the LUMO of $B_{10}H_{13}^+$ showed high surface density located in the B6 position of the cluster, consistent with electrophilic substitution proceeding by the interaction of B6 of $B_{10}H_{13}^+$ with the π-orbital of the aryl ring (HOMO of **44**). As discussed above, this reaction forms two possible intermediates, Int_A (B6 *ipso* to H) and Int_B (B6 *ipso* to isopropyl). The computed energies of Int_A and Int_B (Fig. 4e) indicate that Int_A is 11.0 kcal/mol more stable than Int_B, consistent with the expectation for stabilization of the intermediate by *o* and *p* alkyl groups. The overall reaction from $B_{10}H_{13}^+$ and **44** to the decaborane product, 6-(2,4,6-$^i$Pr$_3$Ph)-*nido*-$B_{10}H_{13}$ (**47**) is highly exergonic by 108.0 kcal/mol, consistent with the extraordinarily high reactivity of the non-classical boron cluster $B_{10}H_{13}^+$. Ion pairing and coordination of water do decrease the extent of exergonicity, but they do not affect the calculated regioselectivity (see Supplementary Figs. 149 and 150).

To examine the steric effect between alkyl groups and the boron cluster in 6-Ar-*nido*-decaborane, 6-(2,4,6-$^i$Pr$_3$Ph)-*nido*-$B_{10}H_{13}$ (**47**) was independently synthesized and characterized by NMR spectroscopy and X-ray diffraction (Fig. 5). Addition of 5 equivalents of HOTf to a mixture of **1** and **44** at 0 °C produced the desired decaborane **47** in 16% isolated yield after 3 h of stirring (Fig. 5a). Slow evaporation of a saturated benzene solution of **47** afforded a single crystal suitable for X-ray diffraction (Fig. 5b). Next, the solid-state structures of **47** and **2**[36] obtained by single crystal X-ray diffraction were compared to investigate the effect of two isopropyl groups versus hydrogen atoms *ortho* to the boron cluster (Fig. 5b, c). A bond distance between the boron cluster and the aryl ring (B6-C1) in **47** is elongated (1.582(2) Å) compared to the corresponding bond distance in **2** (1.555(3) Å). Also, a torsional angle between the B6-B2 plane in the cluster and the aryl plane in **47** (B2-B6-C1-C2 157.3(1)°) is closer to 180° rather than 90°, which means the aryl ring is vertical to the boron cluster, while the corresponding torsional angle in **2** (B2-B6-C1-C7 116.6(2)°) is closer to 90° meaning the phenyl ring is rather planar to the boron cage. The

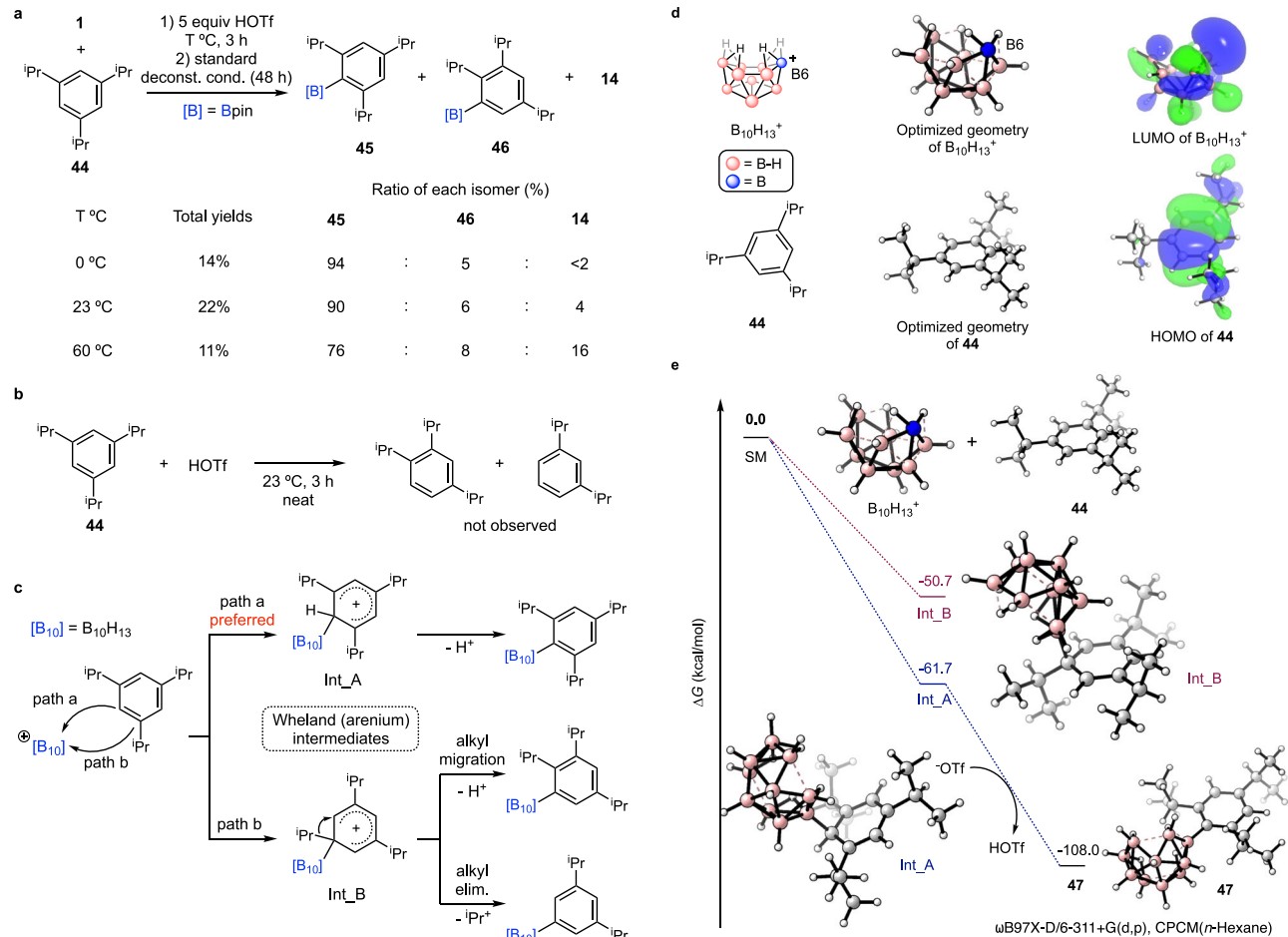

**Fig. 4 | Borylation of 44 and DFT computational studies. a** Borylation of **44** giving three isomers (**45**, **46** and **14**) as the products. **b** Reaction of HOTf and **44**. **c** Two pathways in the regioselective step in borylation of **44**. **d** DFT-optimized geometries of $B_{10}H_{13}^+$ and **44** and visualized LUMO of $B_{10}H_{13}^+$ and HOMO of **44**.

**e** DFT-calculated reaction profile for the formation of **47** from $B_{10}H_{13}^+$ and **44**. All DFT calculations were conducted at the ωB97X-D/6-311+G(d,p), CPCM(n-hexane) level of theory.

observed significant geometric difference in two decaborane compounds, **44** and **2**, can be attributed to the steric effect of the isopropyl groups in **47**. To reduce the steric hindrance by the hydrogen atoms of the boron cluster, one isopropyl group is located above the boron cage and another isopropyl group is situated aside the boron cluster with a slight distortion due to the steric hindrance between the two hydrogen atoms (H2 and H13). This observation clearly shows that the steric hindrance should affect the geometry and stability of Wheland intermediates as well as 6-Ar-*nido*-decaborane compounds, and it supports the existence of the steric effect imposed by the non-classical boron cluster electrophile.

Conventionally, many of the polynuclear boron-rich molecules are thought of as exceedingly stable[54] and thus little attention has been paid toward their potential selective degradation pathways into mononuclear species besides notable examples in partial carborane deboronations[55]. This work showcases how a large polynuclear $B_{10}$-based cluster species can be selectively deconstructed into well-defined mononuclear boron-based small molecule organic compounds. Importantly, this deconstruction chemistry can be done with substituted $B_{10}$-based clusters containing exopolyhedral B–C bonds, leading to the selective destruction of only cluster-based B–B bonds. The selective deconstruction coupled with unique non-classical electrophilic properties of the parent $B_{10}H_{13}^+$ borenium intermediate can be thus leveraged into a straightforward, open flask two-step sequence for arene borylation. This discovery opens up a route towards introducing functional mononuclear boron-based sites into arene

molecules via direct C–H functionalization and further bridges the chemistry of boron clusters and single-site boron-containing small molecules.

## Methods

Caution: (1) triflic acid is a superacid and thus is very reactive, so special caution is required when it is added to any potentially reactive or flammable chemicals, for example, nitroarenes and so on. (2) The parent unfunctionalized decaborane, *nido*-$B_{10}H_{14}$, is known to be neurotoxic.

### Preparation of Cs₂[*closo*-B₁₀H₁₀] (1) from [HNEt₃]₂[B₁₀H₁₀][36]

In a typical experiment, a 20 mL scintillation vial was charged with a magnetic stir bar, 3.22 g (10 mmol) of [HNEt₃]₂[B₁₀H₁₀] and 5 mL distilled $H_2O$ under $N_2$ or aerobic conditions. 3.36 g (20 mmol) of CsOH•H₂O was dissolved in a minimum amount of distilled $H_2O$, and the aqueous CsOH solution was slowly added to the solution of [HNEt₃]₂[B₁₀H₁₀] with vigorous stirring. After 24 h, precipitates from the aqueous solution were filtered using a glass frit, and the collected solid on the glass frit was washed with 2 × 10 mL of EtOH and 3 × 10 mL of Et₂O then dried under vacuum to give 3.2 g (8.3 mmol, 83%) of a white solid identified as **1**.

### Preparation of 6-Tol-*nido*-B₁₀H₁₃ (2) from 1[36]

In a typical experiment, a 20 mL scintillation vial was charged with a magnetic stir bar, 770 mg (2.0 mmol) of **1** and 5 mL toluene under

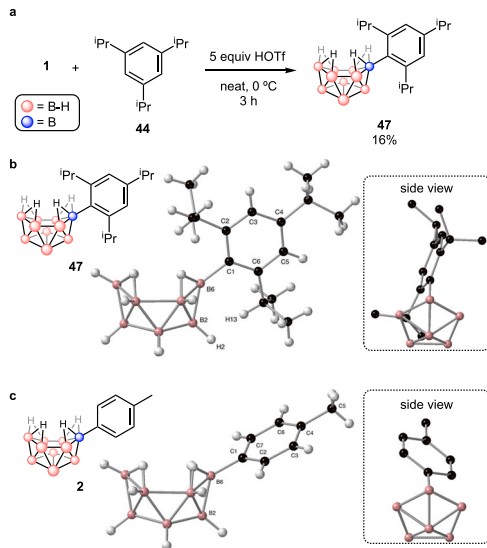

**Fig. 5 | Comparison of solid-state structures of 47 and 2. a** Synthesis of **47**. **b** Solid-state structure of **47**. Selected bond distance (Å) and torsional angle (°): B6-C1 1.5818(15); B2-B6-C1-C2 157.27(11). **c** Solid-state structure of **2** (ref. [36]) for comparison. Selected bond distance (Å) and torsional angle (°): B6-C1 1.555(3); B2-B6-C1-C7 116.6(2). The insets are the side views projecting from the boron cages to the aryl rings.

aerobic conditions. Then, 0.90 mL of HOTf (10.0 mmol, 5.0 equiv) was added to the reaction mixture and the mixture was stirred for 3 h at 23 °C. After 3 h, the volatiles were removed under vacuum and the resulting residue was purified by silica gel flash column chromatography using hexane as an eluent. Fractions were collected and the volatiles were removed under vacuum to give 378 mg (1.78 mmol, 89%) of a white solid identified as **2**.

### General procedure for borylation of alkyl- and halobenzene

In a typical experiment, a 4-mL dram vial was charged with a magnetic stir bar, 154 mg of **1** (0.40 mmol, 1.0 equiv), and 0.5 mL liquid substrates or 5.0 equiv of solid substrates (2.0 mmol) in 0.5 mL hexanes under aerobic conditions. Then, 180 μL of HOTf (2.0 mmol, 5.0 equiv) was added to the reaction mixture and the mixture was stirred for 3 h at 23 °C. The reaction mixture was filtered through a silica plug prepared using a Pasteur pipette packed with ~2 cm length of silica gel. After the filtration, the filter cake was washed with 10 mL hexanes, and the filtrate was collected. Volatiles of the filtrate were removed under vacuum, and the resulting crude mixture was transferred to a 25 mL Schlenk flask charged with a magnetic stir bar, 44 mg of CAN (20 mol%, 0.04 mmol), 4.0 mmol of diol, 580 mg of magnesium sulfate (4.8 mmol, 12.0 equiv) and 10 mL MeCN. The reaction mixture was stirred at 65 °C for 24 h unless otherwise noted. After 24 h, the reaction mixture was filtered through a pad of Celite, and the filtrate was dried under a vacuum to remove volatiles. The crude mixture was purified by silica gel flash column chromatography using hexanes/EtOAc as eluents. Finally, the collected eluents were concentrated under a vacuum to give the desired aryl boronic ester as the product, which was analyzed by NMR spectroscopy.

### Data availability

The crystallographic data generated in this study have been deposited to the Cambridge Crystallographic Data Centre database under accession code CCDC 2189086. Supplementary Data 1 contain the Cartesian coordinates of the structures, and Supplementary Data 2 contain the crystallographic data. The authors declare that all data supporting the findings of this study are available within the paper and its supplementary information files. Data are available from the corresponding author upon request.

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

## Acknowledgements

A.M.S. thanks NIGMS (Grant R35GM124746) for supporting this work. We thank Boron Specialties for a generous donation of [$B_{10}H_{10}$]$^{2-}$ salt precursors. J.W.T. and K.N.H. acknowledge that this work used computational and storage services associated with the Hoffman2 Shared Cluster provided by the UCLA Institute for Digital Research and Education's Research Technology Group.

## Author contributions

S.K. and A.M.S. conceived the idea and co-wrote the manuscript with input from all of the co-authors. S.K. conducted the majority of experiments and Y.A.N. and J.A.M.G. assisted in the experiments. J.W.T. and K.N.H. designed and performed the computational studies. M.G. collected and solved the crystal structure using data from X-ray diffraction. A.M.S. supervised the project.

## Competing interests

The authors declare no competing interests.
