## [Peer Review File · Nature Communications]

Arene C-H Borylation Strategy Enabled by a Non-Classical Boron Cluster-Based ElectrophileREVIEWER COMMENTS

Reviewer #1 (Remarks to the Author):

Aryl boronic esters have proven to be fascinating building blocks due to their versatility for further synthetic transformations. C–H Borylation of arenes represents one of the most practical approaches to constructing these compounds in the atom- and step-economic manner. In this manuscript, Spokoyny and co-workers developed a metal-free strategy for preparation of aryl boronic esters through an electrophilic aromatic substitution induced by cage-opening of Cs₂[closo-B₁₀H₁₀] followed by an unprecedented cage deconstruction of arylated decaboranes promoted by diols. Hawthorne has previously reported triflic acid induced metal-free electrophilic reactions of protonated [closo-B₁₀H₁₀]²⁻ with arenes (ref34-35: J. Am. Chem. Soc. 1992,114, 4427; Inorg. Chem. 2012,51, 9935); Recently, the authors published an elegant work on substitution and cage deconstruction reaction sequence with a sterically unprotected nucleophilic boron cluster reagents to afford boronic acid pinacol esters (ref40: Chem, 2019, 5, 2461). In the present paper, The well-known boron cluster-based electrophile are used in place of nucleophilic boron cluster reagents. There is no new concept, but perfect control of regioselectivity of C–H borylation of sterically hindered alkyl- and haloarenes was achieved and this deconstruction chemistry can be done with substituted B₁₀-based clusters containing exopolyhedral B-C bonds was also demonstrated. The method is very interesting and offers great synthetic potential. The methodology is likely to be of wide interest to those engaged in the preparation of aryl boronic esters and development of boron clusters chemistry. The manuscript is well-written and the conclusions are sufficiently supported by the experimental results. Therefore, I recommends this manuscript to be published in Nature communication after the following minor revisions are addressed:

1. Check the references and the subsequent articles. In the text on page 1 “(EAS)¹²”: the authors reference 12 but reference 12 seems irrelevant to electrophilic aromatic substitution; the corresponding numbers in the reference list should be 10; Also, check the references “directing groups¹²⁻¹³”.
2. Besides arenes, the authors may add more substrates (eg. heteroarenes) and add some discussions on its limitation.
3. Page S2: [TEAH][B₁₀H₁₀] is a too professional term; please avoid using the abbreviation. Procedure for synthesis of [TEAH][B₁₀H₁₀] should be described or related articles should be cited.
4. Page S4: “3.22 g (0.10 mmol) of 1”---should it be “3.22 g (0.10 mmol) of [TEAH][B₁₀H₁₀]”?
5. Page S12: The ¹H NMR data for product 19 is wrong. There is no spectral data of 19 in ref24 (J. Am. Chem. Soc. 2015, 137, 12211). High-resolution mass spectra for compound 19 should be provided.
6. Page S19-20: The numbering of 4,4,5,5-tetramethyl-2-(2,4,6-triisopropylphenyl)-1,3,2-dioxaborolane and 4,4,5,5-tetramethyl-2-(2,3,5-triisopropylphenyl)-1,3,2-dioxaborolane are incorrect; “Preparation of 41 by Borylation”---should it be “Preparation of 42 by Borylation”?

Reviewer #2 (Remarks to the Author):

This report from Spokoyny and Houk reports the activation of a closo borane cluster under highly Bronsted acidic conditions to generate a boron electrophile that effects C-H borylation of arenes, including hindered arenes. This builds on the work of Hawthorne who reported on similar observations as mentioned in the intro. (the authors should also reference Sivaev’s work using another borane cluster to perform electrophilic C-H borylation see: - C–H Bond Activation of Arenes by [8,8’-μ-I-3,3’-Co(1,2-C₂B₉H₁₀)₂] in the Presence of Sterically Hindered Lewis Bases | Organometallics (acs.org).

The major new elements of this work therefore are the expanded scoping study (relative to Hawthorne's and Sivaev's work) and the ability to deconstruct the Aryl-Bcluster product into ArylBpin, though cluster deconstruction has been reported for a related borane cluster recently by the same group (ref 40). Given the highly inefficient nature of the process (excess HOTf required, large amounts of boron based waste by-products, 5 equivalents of arene required) the value of this process in my opinion has to be judged on two things: (i) the ability to access sterically hindered borylated product directly from the C-H arene as opposed to previous work where the aryl-bromide (for example) had to be formed as an intermediate before lithiation/borylation; (ii) the ability to perform the borylations under air without glovebox etc. If more work is performed on both of these then it is possible this work would become suitable for Nat Comm

For the former some work needs to be performed to determine if this procedure is really useful, as simple alkylarenes and haloarenes have been previously borylated using simpler boron electrophiles (refs 27-29) e.g. are heavier halides tolerated (such as bromide?) and are any other acid stable functional groups tolerated (CF₃)? Currently it is just alkyl, fluoride and chloride that are reported. In terms of sterically hindered systems, these are novel for C-H borylation, but can higher yields e.g. of 23, 35, 42, be produced? the highest shown in Fig. 5 for 42 is 22%.

In terms of increased scope/ borylation reactivity the authors should look at using the commercially available stronger acid HNtF₂. If there is a B---anion contact in the reactive species than the use of NTf₂ in place of OTf may enhance reactivity (As NTf₂ is generally less coordinating), increase yields (and possibly lower the amounts of H⁺ required). Indeed, what happens if lower than 5 equiv. of HOTf is used?

The air stability of this process is notable, though is this due to the low concentration of water present in 0.5 ml of hexane / neat arene when performing the reaction in sealed vial conditions? Is it really water tolerant? Linked to this- regarding the calculations, these seem overly downhill in my opinion. The authors make the statement "Ion pairing will decrease this exergonicity to some extent, although we did not compute that value." I think this is an understatement, the starting material will not be the free B₁₀H₁₃ cation in solution as the authors are aware. Something will be interacting with the empty orbital projecting out from the cluster core (LUMO). Therefore, it will be v. interesting to see on modelling if the B-OTf or a B—OH₂ adduct is the energy minimum (i.e. is water of triflate more coordinating to this boron cation). If it is the latter than this has implications for true water tolerance. Starting from whichever of these SM is lower in energy is more appropriate to give more accurate energetics for this transformation.

Minor comments

In the SI "one dram vial" what is one dram in ml?

For some more of the ArylBPin products some ¹¹B NMR spectra should be shown in the SI to confirm the absence of boron based impurities that may not be visible in the ¹H NMR spectrum (e.g. B-OH or B-H based impurities). I think only ¹¹B for 17 and 30 are shown currently in the SI for the ArylBPins.

Reviewer #3 (Remarks to the Author):

In this work entitled "Arene C-H Borylation Strategy Enabled by a Non-Classical Boron Cluster-Based Electrophile" by A. Spokoyny and coworkers a new, elegant method for borylation of aromatic rings using nido-B₁₀H₁₃ cation is described. Using this method the authors showed borylation of very challenging aryl rings bearing very bulky substituents and electron withdrawing group (F, Cl). Some of the aryl rings that were borylated in this paper were not previously accessible using transition metal-free methods. The initial products (6-Ar-nido-B₁₀H₁₃) were converted to the more common Bpin derivatives by treatment with excess of pinacol and heating. This transformation was shown to be not limited to Bpin, but also other boronic esters groups were prepared.

· These results are very interesting and could become useful in synthetic organic chemistry, especially the more challenging cases are of interest.

· Since transition metal-free methods are mostly based on the electrophilic aromatic substitution most of them rely on electron rich aromatic rings. This method uses the highly electrophilic B-center in nido-B10H13 and thus works on aromatic ring that are not substituted with highly electron donating groups. Surprisingly, this method also works on highly sterically congested systems, which is very interesting.

· The claims and conclusions of this work are well supported by the presented experimental and theoretical data.

· The methodology is sound and indeed meets the expected standards

· All details are provided and the work should be reproducible based on the provided data.

Overall this manuscript presents very interesting chemistry, written in a good and easily readable way. I believe that this work definitely merits publication in Nature Communications. Few minor things to be corrected/considered by the authors.

1. I think it'd be very useful to run a "background" reaction between HBpin or other hydroboranes (9-BBN would be very interesting), HOTf and aryl ring of a choice to make sure the borylation does not occur directly with these boranes. I don't remember whether this was ever reported without stabilizing ligand at B-center.

2. Although the transformation of nido-B10H13 to Bpin is very interesting, and most of organic chemists are more used to Bpin and other common boryl substituents, I wonder if nido-B10H13 group can be directly used in coupling reactions. If yes this would be really interesting and important, since this will save the replacement to Bpin reaction.

Few small corrections:

Page 7, 2nd paragraph in the end: "Chlorobenzenes generally showed lower.... patterns (29-35)" I think the numbers are wrong, should be 30, 32, 35.

Page 9, 2nd paragraph: typo in the sentence starting with: "The computed energies of Int_A and Int_B..." is 11 kcal/lower than Int_B by 11 kcal/mol

In Data availability section: Crystallographic data for 44 is missing.

The provided comments are in black, and our responses to those comments are in blue.

Reviewer #1 (Remarks to the Author):

Aryl boronic esters have proven to be fascinating building blocks due to their versatility for further synthetic transformations. C–H Borylation of arenes represents one of the most practical approaches to constructing these compounds in the atom- and step-economic manner. In this manuscript, Spokoyny and co-workers developed a metal-free strategy for preparation of aryl boronic esters through a electrophilic aromatic substitution induced by cage-opening of Cs₂[closo-B₁₀H₁₀] followed by an unprecedented cage deconstruction of arylated decaboranes promoted by diols.

Hawthorne has previously reported triflic acid induced metal-free electrophilic reactions of protonated [closo-B₁₀H₁₀]²⁻ with arenes (ref34-35: J. Am. Chem. Soc. 1992,114, 4427; Inorg. Chem. 2012,51, 9935); Recently, the authors published an elegant work on substitution and cage deconstruction reaction sequence with a sterically unprotected nucleophilic boron cluster reagents to afford boronic acid pinacol esters (ref40: Chem, 2019, 5, 2461). In the present paper, The well-known boron cluster-based electrophile are used in place of nucleophilic boron cluster reagents. There is no new concept, but perfect control of regioselectivity of C–H borylation of sterically hindered alkyl- and haloarenes was achieved and this deconstruction chemistry can be done with substituted B₁₀-based clusters containing exopolyhedral B-C bonds was also demonstrated. The method is very interesting and offers great synthetic potential. The methodology is likely to be of wide interest to those engaged in the preparation of aryl boronic esters and development of boron clusters chemistry. The manuscript is well-written and the conclusions are sufficiently supported by the experimental results. Therefore, I recommends this manuscript to be published in Nature communication after the following minor revisions are addressed:

1. Check the references and the subsequent articles. In the text on page 1 “(EAS)¹²”: the authors reference 12 but reference 12 seems irrelevant to electrophilic aromatic substitution; the corresponding numbers in the reference list should be 10; Also, check the references “directing groups¹²⁻¹³ “. We thank the reviewer for the correction. “EAS¹²” and “directing groups¹²⁻¹³” have been changed to “EAS¹⁰” and “directing groups¹¹⁻¹³”, respectively.

2. Besides arenes, the authors may add more substrates (eg. heteroarenes) and add some discssions on its limitation. The reactivity of B₁₀H₁₀²⁻ under acidic conditions is inherently not suitable for borylation of heteroarenes due to the use of HOTf which results in rapid formation of protonated heterocycle-based salts. We have added a statement about the limitations in the revised manuscript. However, we have discovered an alternative synthetic route to make ArBpin using B₁₀H₁₀²⁻ as the boron source which has been added in the revised manuscript. Specifically, thermal arylation of [TBA]₂[B₁₀H₁₀] (TBA = ⁿBu₄N) in the presence of 1,3-dibromobenzene afforded polyarylated decaborate [TBA]₂[B₁₀Ar_nH_{10-n}] (Ar = 3,5-Br₂Ph, n = 6-7) which undergoes oxidative deconstruction of the boron cage to yield the corresponding ArBpin product. This process is

reminiscent of the oxidative deconstruction of peralkylated B₆H₆-based clusters which we have previously reported.

3. Page S2: [TEAH][B₁₀H₁₀] is a too professional term; please avoid using the abbreviation. Procedure for synthesis of [TEAH][B₁₀H₁₀] should be described or related articles should be cited. We agree with the reviewer's comment. "[TEAH]"s in the SI have been revised to "[HNEt₃]₂" accordingly.

4. Page S4: "3.22 g (0.10 mmol) of 1"---should it be "3.22 g (0.10 mmol) of [TEAH] [B₁₀H₁₀]"? We thank the reviewer for the correction. "1" has been changed to "[HNEt₃]₂[B₁₀H₁₀]" and molarities of each reagent have been corrected.

5. Page S12: The ¹H NMR data for product 19 is wrong. There is no spectral data of 19 in ref24 (J. Am. Chem. Soc. 2015, 137, 12211). High-resolution mass spectra for compound 19 should be provided. Some of the aromatic peaks of 19 were overlapped with the residual solvent peak. This region has been expanded and adequate integrations are provided in the revised supporting information. Also, the HR-MS (DART) of 19 as well as ¹¹B NMR spectrum have been added in the SI.

6. Page S19-20: The numbering of 4,4,5,5-tetramethyl-2-(2,4,6-triisopropylphenyl)-1,3,2-dioxaborolane and 4,4,5,5-tetramethyl-2-(2,3,5-triisopropylphenyl)-1,3,2-dioxaborolane are incorrect; "Preparation of 41 by Borylation"---should it be "Preparation of 42 by Borylation"? We thank the reviewer for finding the typo. 41 has been changed to 45 (not 42 due to addition of new compounds 41-43) accordingly.

Reviewer #2 (Remarks to the Author):

This report from Spokoyny and Houk reports the activation of a closo borane cluster under highly Bronsted acidic conditions to generate a boron electrophile that effects C-H borylation of arenes, including hindered arenes. This builds on the work of Hawthorne who reported on similar observations as mentioned in the intro. (the authors should also reference Sivaev's work using another borane cluster to perform electrophilic C-H borylation see:- C-H Bond Activation of Arenes by [8,8'-μ-I-3,3'-Co(1,2-C₂B₉H₁₀)₂] in the Presence of Sterically Hindered Lewis Bases I Organometallics (acs.org). The reference has been added.

The major new elements of this work therefore are the expanded scoping study (relative to Hawthorne's and Sivaev's work) and the ability to deconstruct the Aryl-Bcluster product into ArylBpin, though cluster deconstruction has been reported for a related borane cluster recently by the same group (ref 40). Given the highly inefficient nature of the process (excess HOTf required, large amounts of boron based waste by-products, 5 equivalents of arene required) the value of this process in my opinion has to be judged on two things: (i) the ability to access sterically hindered borylated product directly from the C-H arene as opposed to previous work where the aryl-bromide (for example) had to be formed as an intermediate before lithiation/borylation; (ii) the ability to perform the borylations under air without glovebox etc. If more work is performed on both of these then it is possible this work would become suitable for Nat Comm In the revised manuscript, (i) we have added an additional experimental procedure showcasing a strategy amenable to the borylation of an arylbromide substrate. Specifically, thermal arylation of [TBA]₂[B₁₀H₁₀] in the presence of 1,3-dibromobenzene afforded polyarylated decaborate [TBA]₂[B₁₀Ar_nH_{10-n}] (Ar = 3,5-Br₂Ph, n = 6-7) which undergoes oxidative deconstruction

of the boron cage to yield the corresponding ArBpin product. (ii) The reaction can be fully performed under aerobic conditions (with literally 'open'-flask conditions) and air-free conditions have never been employed to handle any of compounds in this study. We hope our additional experiments and comments clarify and satisfy the reviewer's critique.

For the former some work needs to be performed to determine if this procedure is really useful, as simple alkylarenes and haloarenes have been previously borylated using simpler boron electrophiles (refs 27-29) e.g. are heavier halides tolerated (such as bromide?) and are any other acid stable functional groups tolerated (CF₃)? Currently it is just alkyl, fluoride and chloride that are reported. In terms of sterically hindered systems, these are novel for C-H borylation, but can higher yields e.g. of 23, 35, 42, be produced? the highest shown in Fig. 5 for 42 is 22%. Our current reaction conditions of the sequential cage-opening and deconstruction reactions cannot perform for aryl bromides and CF₃-substituted arenes due to intrinsic limitation of electrophilic aromatic substitution. Although the non-classical boron cluster-based electrophile is very electrophilic, the deleterious side reaction, formation of B₁₀H₁₃OTf from [B₁₀H₁₃]⁺ and [OTf]⁻, hampers successful EAS of those substrates. Despite these methodological limitations, we believe this work nevertheless represents a conceptual departure from the existing *status quo* in the field. Furthermore, considering that we have introduced a proof-of-concept approach centered on a thermal borylation strategy that tolerates more sensitive aryl-based substrates (see above), we think that the reviewer's concerns should be substantially mitigated.

In terms of increased scope/ borylation reactivity the authors should look at using the commercially available stronger acid HNTf₂. If there is a B---anion contact in the reactive species than the use of NTf₂ in place of OTf may enhance reactivity (As NTf₂ is generally less coordinating), increase yields (and possibly lower the amounts of H⁺ required). We thank the reviewer for the great suggestion. With chloromesitylene as a substrate, we have tested HNTf₂ as the proton source instead of HOTf, however, we have obtained only a trace amount of the desired product, while the original conditions utilizing HOTf gave 39% isolated yield. We believe that the following rationale can be used to explain these differences. (i) HOTf, a liquid, can be added to a suspension of Cs₂B₁₀H₁₀ that is insoluble in aromatic solvents (as substrates) to react in a heterogeneous manner. However, HNTf₂ is a solid and is not well soluble in aromatic solvents, either. It makes the overall reaction kinetics poorer than the reaction using HOTf and the cage-opening reaction does not take place that well. (ii) In addition, the resulting CsNTf₂ salt is a rocky aggregate which stops stirring completely during the reaction while CsOTf is a soft, powder-like solid which does not make any issues. Although we agree with the reviewer's point that HNTf₂ is likely to be better than HOTf, some unpredictable issues make it difficult being used as a proton source in this reaction.

Indeed, what happens if lower than 5 equiv. of HOTf is used? The cage-opening of B₁₀H₁₀²⁻ was incomplete and it decreased the reaction yields significantly.

The air stability of this process is notable, though is this due to the low concentration of water present in 0.5 ml of hexane / neat arene when performing the reaction in sealed vial conditions? Is it really water tolerant? The reactions were fully performed under aerobic conditions without using any Schlenk techniques or glove box. We used all the glassware, reagents and apparatus (e.g. needles and plastic syringes) which have been stored

under air. The majority of performed reactions was conducted during the summer when humidity in southern California is high. Based on all of these considerations operationally, this reaction is not hampered by residual moisture. Linked to this- regarding the calculations, these seem overly downhill in my opinion. The authors make the statement “Ion pairing will decrease this exergonicity to some extent, although we did not compute that value.” I think this is an understatement, the starting material will not be the free B₁₀H₁₃ cation in solution as the authors are aware. Something will be interacting with the empty orbital projecting out from the cluster core (LUMO). Therefore, it will be v. interesting to see on modelling if the B-OTf or a B—OH₂ adduct is the energy minimum (i.e. is water of triflate more coordinating to this boron cation). If it is the latter than this has implications for true water tolerance. Starting from whichever of these SM is lower in energy is more appropriate to give more accurate energetics for this transformation.

As the reviewer suggested, we have calculated the energy profiles including B-OTf and B-OH₂⁺ species as shown below.

The reaction of B₁₀H₁₃OTf with 41 to produce 44 as the product was revealed as a thermodynamically uphill reaction, which shows that B₁₀H₁₃OTf is not an intermediate for this reaction. Actually, B₁₀H₁₃OTf is the major side product where EAS by the cage-opening of B₁₀H₁₀²⁻ does not work successfully. Based on these considerations, formation of B₁₀H₁₃OTf is likely a competing side reaction.

On the other hand, the reaction of B₁₀H₁₃OH₂⁺ and 41 was found out to be thermodynamically downhill, which supports this reaction can be water tolerant.

We have added these results in the SI. We thank the reviewer for the great suggestion to improve our manuscript.

Minor comments In the SI “one dram vial” what is one dram in ml? We have revised “one dram vial” to “4-mL dram vial” throughout the SI. We thank the reviewer for the correction.

For some more of the ArylBPins products some ¹¹B NMR spectra should be shown in the SI to confirm the absence of boron based impurities that may not be visible in the ¹H NMR spectrum (e.g. B-OH or B-H based impurities). I think only ¹¹B for 17 and 30 are shown currently in the SI for the ArylBPins. We have added all ¹¹B NMR spectra of the reported compounds in the SI.

Reviewer #3 (Remarks to the Author):

In this work entitled "Arene C-H Borylation Strategy Enabled by a Non-Classical Boron Cluster-Based Electrophile" by A. Spokoyny and coworkers a new, elegant method for borylation of aromatic rings using nido-B₁₀H₁₃ cation is described. Using this method the authors showed borylation of very challenging aryl rings bearing very bulky substituents and electron withdrawing group (F, Cl). Some of the aryl rings that were borylated in this paper were not previously accessible using transition metal-free methods. The initial products (6-Ar-nido-B₁₀H₁₃) were converted to the more common Bpin derivatives by treatment with excess of pinacol and heating. This transformation was shown to be not limited to Bpin, but also other boronic esters groups were prepared.

· These results are very interesting and could become useful in synthetic organic chemistry, especially the more challenging cases are of interest.

· Since transition metal-free methods are mostly based on the electrophilic aromatic substitution most of them rely on electron rich aromatic rings. This method uses the highly electrophilic B-center in nido-B₁₀H₁₃ and thus works on aromatic ring that are not substituted with highly electron donating groups. Surprisingly, this method also works on highly sterically congested systems, which is very interesting.

· The claims and conclusions of this work are well supported by the presented experimental and theoretical data.

· The methodology is sound and indeed meets the expected standards

· All details are provided and the work should be reproducible based on the provided data.

Overall this manuscript presents very interesting chemistry, written in a good and easily readable way. I believe that this work definitely merits publication in Nature Communications. Few minor things to be corrected/considered by the authors.

1. I think it'd be very useful to run a "background" reaction between HBpin or other hydroboranes (9-BBN would be very interesting), HOTf and aryl ring of a choice to make sure the borylation does not occur directly with these boranes. I don't remember whether this was ever reported without stabilizing ligand at B-center. We thank the reviewer for the suggestion. Accordingly, we have tested the reaction of HOTf and HBpin in mesitylene as a solvent (and a substrate), but it did not produce any detectable amount of MesBpin. The reaction was extremely exothermic and evolution of H₂ was observed, but an orange solid (presumably Bpin(OTf)) was formed.

2. Although the transformation of nido-B₁₀H₁₃ to Bpin is very interesting, and most of organic chemists are more used to Bpin and other common boryl substituents, I wonder if nido-B₁₀H₁₃ group can be directly used in coupling reactions. If yes this would be really interesting and important, since this will save the replacement to Bpin reaction. We thank the reviewer for raising another interesting point. nido-B₁₀H₁₃ is not appropriate as a direct cross-coupling partner due to lack of proper LUMOs essential for transmetalation unlike common aryl boronic esters. However, this also implies that nido-B₁₀H₁₃ can be utilized as a protecting group for iterative cross-coupling reactions similar to the widely used MIDA. (MIDA = N-methyliminodiacetic acid)

Few small corrections: Page 7, 2nd paragraph in the end: "Chlorobenzenes generally showed lower.... patterns (29-35)" I think the numbers are wrong, should be 30, 32, 35. It has been revised accordingly.

Page 9, 2nd paragraph: typo in the sentence starting with: "The computed energies of Int_A and Int_B..." is 11 kcal/lower than Int_B by 11 kcal/mol "by 11 kcal/mol" has been removed.

In Data availability section: Crystallographic data for 44 is missing. The typo "43" was corrected to "47".

REVIEWERS' COMMENTS

Reviewer #2 (Remarks to the Author):

I thank the authors for carefully considering my (and the other reviewer's comments). The additional experiments, calculations and the detailed response in the rebuttal letter have covered all my suggestions/ requested clarifications sufficiently. I now support the publication of this nice work, essentially as is.

Minor comments

The authors should add a footnote somewhere that during the review of this work another catalytic electrophilic borylation process applicable to hindered arenes was reported in JACS. See: Borenium-Ion-Catalyzed C–H Borylation of Arenes (acs.org)